# Increased Adipose Tissue Expression of Interferon Regulatory Factor (IRF)-5 in Obesity: Association with Metabolic Inflammation

**DOI:** 10.3390/cells8111418

**Published:** 2019-11-11

**Authors:** Sardar Sindhu, Reeby Thomas, Shihab Kochumon, Ajit Wilson, Mohamed Abu-Farha, Abdullah Bennakhi, Fahd Al-Mulla, Rasheed Ahmad

**Affiliations:** 1Animal & Imaging Core Facility, Dasman Diabetes Institute (DDI), Al-Soor Street, P.O. Box 1180, Dasman 15462, Kuwait; 2Department of Microbiology & Immunology, Dasman Diabetes Institute (DDI), Al-Soor Street, P.O. Box 1180, Dasman 15462, Kuwait; reeby.thomas@dasmaninstitute.org (R.T.); shihab.kochumon@dasmaninstitute.org (S.K.); ajit.wilson@dasmaninstitute.org (A.W.); 3Department of Biochemistry and Molecular Biology, Dasman Diabetes Institute (DDI), Al-Soor Street, P.O. Box 1180, Dasman 15462, Kuwait; mohamed.abufarha@dasmaninstitute.org; 4Medical Division, Dasman Diabetes Institute (DDI), Al-Soor Street, P.O. Box 1180, Dasman 15462, Kuwait; abdullah.bennakhi@dasmaninstitute.org; 5Department of Genetics & Bioinformatics, Dasman Diabetes Institute (DDI), Al-Soor Street, P.O. Box 1180, Dasman 15462, Kuwait; fahd.almulla@dasmaninstitute.org

**Keywords:** interferon regulatory factor-5, obesity, metabolic inflammation, adipose tissue

## Abstract

Interferon regulatory factor (IRF)-5 is known to be involved in M1 macrophage polarization, however, changes in the adipose expression of IRF5 in obesity and their relationship with the local expression of proinflammatory cytokines/chemokines are unknown. Therefore, IRF5 gene expression was determined in the subcutaneous adipose tissue samples from 53 non-diabetic individuals (6 lean, 18 overweight, and 29 obese), using real-time RT-PCR. IRF5 protein expression was also assessed using immunohistochemistry and/or confocal microscopy. Adipose gene expression of signature immune metabolic markers was also determined and compared with adipose IRF5 gene expression. Systemic levels of C-reactive protein and adiponectin were measured by ELISA. The data show that adipose IRF5 gene (P = 0.008) and protein (P = 0.004) expression was upregulated in obese compared with lean individuals. IRF5 expression changes correlated positively with body mass index (BMI; r = 0.37/P = 0.008) and body fat percentage (r = 0.51/P = 0.0004). In obese, IRF5 changes associated positively with HbA1c (r = 0.41/P = 0.02). A good agreement was found between gene and protein expression of IRF5 in obese subjects (r = 0.65/P = 0.001). IRF5 gene expression associated positively with adipose inflammatory signatures including local expression of TNF-α, IL-6, CXCL8, CCL-2/5, IL-1β, IL-18, CXCL-9/10, CCL7, CCR-1/2/5, TLR-2/7/8/9, IRF3, MyD88, IRAK-1, and inflammatory macrophage markers (P < 0.05). Interestingly, IRF5 gene expression correlated positively with CRP (r = 0.37, P = 0.03) and negatively with adiponectin levels (r = −0.43, P = 0.009). In conclusion, elevated adipose IRF5 expression in obesity concurs with the typical inflammatory signatures, locally and systemically. Hence, the IRF5 upregulation may represent a novel adipose tissue marker for metabolic inflammation.

## 1. Introduction

Obesity and related comorbidities, such as hypertension, cardiovascular disease, insulin resistance, and type-2 diabetes (T2D), are associated with the presence of chronic low-grade inflammation called metabolic inflammation or metaflammation. Obese individuals show both the subcutaneous and intra-abdominal (visceral) fat deposition and the changes occurring in the expanding adipose tissue compartment are pivotal to metabolic inflammation and development of insulin resistance. Adipose tissue is recognized as both energy storage and active endocrine organ. Whereas, small adipocytes in lean individuals are linked to metabolic homeostasis, the enlarged adipocytes in obese individuals recruit activated macrophages as well as secrete adipokines [1]. Proinflammatory cytokines/chemokines secreted by the resident macrophages and adipokines secreted by the enlarged adipocytes act cooperatively to promote the adipose inflammation and lead to insulin resistance [2].

Regarding proinflammatory cytokines/chemokines that are considered as critical biomarkers of adipose inflammation, TNF-α is a potent proinflammatory cytokine which is secreted by macrophages and adipocytes, especially by visceral adipose tissue. This cytokine is directly involved in insulin and glucose metabolism, induces insulin resistance and stimulates lipolysis [3]. Both human and mouse model studies have reported the increased TNF-α levels in obesity and association with BMI [4,5,6]. IL-6 is another proinflammatory cytokine which is predominantly secreted by adipocytes, in addition to macrophages, T cells, endothelial cells, fibroblasts, and skeletal muscle. IL-6 acts as both a proinflammatory cytokine and an anti-inflammatory myokine. IL-6 affects glucose metabolism and increased circulatory levels of IL-6 in obese and/or T2D patients have been documented; however, its role in the development of insulin resistance remains controversial [7,8]. Other proinflammatory cytokines that may be elevated in obesity/T2D and contribute to metaflammation may include IL-1β, IL-18, and IL-23 [9,10,11]. Chemotactic cytokines, also called chemokines, are small proteins that signal by activating seven-transmembrane-domain receptors, and are classified into main subfamilies as CXC, CC, CX3C, and XC chemokines. The main role of chemokines is to act as chemoattractants and direct the migration of leukocytes i.e., induction of directed chemotaxis. The signature metaflammatory chemokines include CXCL-1/5/8/9, CCL-2/3/4/5/7, and CX3CL1 [12,13,14,15].

The interferon regulatory factors (IRFs) family of transcription factors plays a critical role in the regulation of immunity and induction of type I interferons (IFN-α/β) [16]. IRFs comprise a family of 9 transcription factors (IRF1-9) that are involved in immunoregulation and immune cell differentiation by Toll-like and other pattern recognition receptors. IRFs are now emerging as transcriptional regulators of adipogenesis [17]. Notably, IRF5 has been implicated with macrophage polarization toward an inflammatory M1 phenotype as well as adipose deposition and insulin sensitivity in obesity [18]. The changes in IRF5 expression and their relationship with inflammatory markers in the adipose tissue in obesity remain unclear. Herein, we present the data showing increased adipose tissue expression of IRF5 in overweight/obese individuals and its positive correlation with critical inflammatory markers such as proinflammatory cytokines/chemokines, pattern recognition markers, and M1 macrophage markers.

## 2. Materials and Methods

### 2.1. Study Population

A total of 53 non-diabetic individuals, aged from 24–71 years were recruited in the study at the Dasman Diabetes Institute, Kuwait. The individuals suffering from serious illnesses such as disease of the lung, heart, kidney, liver, hematologic disorders, pregnancy, immune dysfunction, diabetes (types 1 and 2), or malignancy were excluded from the study. The participants were grouped as lean (<25 kg/m^2^), overweight (25–30 kg/m^2^), and obese (>30 kg/m^2^) according to their body mass index (BMI). Co-morbid conditions included hypertension (4) and hyperlipidemia (3). The clinical characteristics of the study participants are shown in Table 1. All individuals were recruited in the study following their written informed consent and the study was approved by the ethics committee of Dasman Diabetes Institute, Kuwait (Ref. RA2015-027), which comprises of members from other scientific research institutions at the national level as well as a legal expert, an ethicist, and a member of community from the non-scientific background. The institutional ethics committee follows the current international guidelines and ethical principles for medical research involving human subjects, as per the WMA Declaration of Helsinki (DoH-Oct2013).

### 2.2. Anthropometric and Physio-Clinical Measurements

Anthropometric and physio-clinical measurements were made including body weight, height, waist circumference, and systolic and diastolic blood pressure. Body weight was measured with portable electronic weighing scale and height was measured by using height measuring bars. Waist circumference was measured by using constant tension tape at the end of a normal expiration with arms relaxed at sides. Body composition analyzer (IOI353, South Korea) was used to measure whole body composition including the body fat percentage, soft lean mass, and total body water. Blood pressure was measured with digital automatic sphygmomanometer (Omron HEM-907XL Omron HEM-907XL, Omron Healthcare Inc., Lake Forest, IL, USA). Three consecutive blood pressure readings, 5–10 min rest apart, were obtained. The following formula was used to calculate the BMI: BMI = Body weight (Kg) / Height (m^2^). Peripheral blood samples were collected from the study participants after overnight (10 h minimum) fasting and samples were analyzed to determine the fasting plasma glucose, glycated hemoglobin (HbA1c), fasting serum insulin, and serum lipids levels. Glucose and lipids were detected by using Siemens dimension RXL chemistry analyzer (Diamond Diagnostics, Holliston, MA, USA). HbA1c was detected with Variant™ device (BioRad, Hercules, CA, USA). To obtain plasma, anticoagulated blood was centrifuged at 1200× *g* for 10 min and collected plasma was aliquoted and stored at −80 °C until use. Plasma triglycerides were determined using commercial kit (Intra-assay CV% = 0.93; Inter-assay CV% = 3.05) (Chema Diagnostica, Monsano, Italy). Levels of plasma C-reactive protein (CRP) and adiponectin were determined by using commercial kits (Cat. # DY1707 Human CRP DuoSet ELISA kit and Cat. # DRP300 Human Total Adipokine/Acrp30 Quantikine ELISA kit, R&D systems, USA). All assays were performed following the manufacturers’ instructions.

### 2.3. Collection of Subcutaneous Adipose Tissue

The abdominal subcutaneous fat tissue samples, about 0.5 g each, were collected by standard biopsy method lateral to the umbilicus as mentioned [7]. Briefly, the periumbilical area was disinfected by swabbing with alcohol and locally anesthetized by injecting 2 mL of 2% lidocaine. A small skin incision (0.5 cm) was made to collect the subcutaneous adipose tissue. Fat tissue was further incised into small fragments, rinsed in cold phosphate buffered saline (PBS), fixed in 4% paraformaldehyde for 24 h and finally embedded in paraffin. At the same time, freshly collected fat tissue samples (50–100 mg size) were also preserved in RNA later and stored at −80 °C until use.

### 2.4. Real-Time Reverse-Transcription Polymerase Chain Reaction (RT-PCR)

To determine gene expression, total RNA was purified from the adipose tissue following the manufacturer’s instructions (RNeasy kit, Qiagen, Valencia, CA, USA) as described [7]. The isolated RNA was quantified using Epoch™ Spectrophotometer (BioTek, Winooski, VT, USA) and RNA quality was evaluated by formaldehyde-agarose gel electrophoresis. One microgram of each RNA sample was reverse transcribed into cDNA using random hexamer primers and TaqMan RT reagents (High Capacity, cDNA RT Kit; Applied Biosystems, Foster City, CA, USA). Then, cDNA (50 ng) was amplified by using TaqMan® Gene Expression MasterMix (Applied Biosystems) together with target gene-specific 20× TaqMan Gene Expression Assays (Applied Biosystems) containing forward/reverse primers (Appendix A) and target-specific TaqMan® minor groove binder (MGB) probe labeled with 6-fluorescein amidite (FAM) dye at 5’ and with non-fluorescent quencher (NFQ)-MGB at 3’ end of the probe for 40 cycles of PCR amplification using 7500 Fast Real-Time PCR System (Applied Biosystems, CA, USA). Each thermal cycle included heating at 95 °C for 15 s for denaturation, then heating at 60 °C for 1 min for annealing/extension, followed by heating at 50 °C for 2 min for uracil DNA glycosylase activation and later, heating at 95 °C for 10 min for AmpliTaq Gold enzyme activation. The expression of glyceraldehyde 3-phosphate dehydrogenase (GAPDH) was used as internal control to normalize differences in individual samples compared to control sample (lean adipose tissue). Target gene expression (relative mRNA expression) was calculated using the 2^−ΔΔCt^ method and was expressed as fold change (mean ± SEM) over the average GAPDH expression taken as one.

### 2.5. Immunohistochemistry (IHC)

For detecting protein expression by IHC, paraffin-embedded 4 µm-thick adipose tissue sections were processed as described elsewhere [7]. Briefly, samples were treated overnight at room temperature with rabbit anti-human primary antibodies against IRF5 (diluted 1:400, Abcam® ab140593), TNF-α (diluted 1:800, Abcam® ab9635), IL-6 (diluted 1:400, Abcam® ab154367), CXCL8 (diluted 1:200, Abcam® ab106350), CCL2 (diluted 1:400, Abcam® ab9669), and CCL5 (diluted 1:200, Abcam® ab9679). After three washes with PBS-Tween, samples were incubated for 1 h with secondary antibody i.e., goat anti-rabbit Alexa Fluor 594-conjugated Ab (diluted 1:200, Abcam® ab150088) and color was developed by using 3,3ʹ-diaminobenzidine (DAB) chromogenic substrate. Specimens were washed at least thrice, counterstained, dehydrated, cleared, and mounted as described [19]. For analysis, digital photomicrographs (20×; Olympus BX51 Microscope, Japan) of fat tissue samples were used to quantify the staining in three different regions delineated by using ImageScope software (Aperio, Vista, CA, USA). Staining intensity was quantified by using Aperio-positive pixel count algorithm (version 9) and was expressed as arbitrary units (AU). The number of positive pixels was normalized in reference to the number of total pixels (positive and negative). Color/thresholds were set to detect the immunostaining as positive and the background as negative pixels. All samples were analyzed using the same parameters and the resulting color markup of analysis was confirmed for each sample.

### 2.6. Confocal Microscopy

For confocal microscopy, formalin-fixed, paraffin-embedded 8 μm thick fat tissue samples were immunolabeled using a similar protocol as used for IHC. After antigen retrieval and blocking, samples were incubated at room temperature overnight with primary antibody i.e., mouse anti-human IRF5 mAb (diluted 1:50, Abcam® ab140593). After washing thrice with PBS-Tween, samples were incubated for 1 h with secondary antibody i.e., goat anti-mouse Alexa Fluor 647-conjugated Ab (diluted 1:200, Abcam® ab150115) and washed at least three times. Samples were counterstained with 4ʹ,6-diamidino-2-phenylindole (DAPI) (Vectashield, Vector Laboratories, H1500) and mounted. Confocal images were obtained by using inverted Zeiss LSM710 spectral confocal microscope (Carl Zeiss, Gottingen, Germany) and EC Plan-Neofluar 40×/1.30 oil DIC M27 objective lens. After exciting samples with 543 nm HeNe laser and 405 nm line of an argon ion laser, optimized emission detection bandwidths were configured by using Zeiss Zen 2010 control software.

### 2.7. Statistical Analysis

The data obtained were expressed as mean ± SEM values. Analysis of variance (Dunnett’s test) was used to compare overweight and obese groups with the lean control group. For statistical analysis of the data as well as preparation of graphics, GraphPad Prism software (version 6.05; San Diego, CA, USA) was used. All p-values ≤ 0.05 were considered as statistically significant.

## 3. Results

### 3.1. Increased Adipose IRF-5 Expression in Obesity Correlates with BMI, Body Fat Percentage, Age, and HbA1c

In metabolic disorders, immunometabolic changes occur in the white adipose tissue which serves as both a fat storage and endocrine organ. Therefore, we asked if the IRF5 adipose tissue gene expression was modulated in overweight/obese. To this end, our data show that IRF5 gene expression was significantly upregulated in overweight (P = 0.05) and obese (P = 0.008) individuals compared with lean (Figure 1A). As expected, IRF5 protein expression, as determined by IHC, was also higher in overweight (P = 0.18) and obese (P = 0.004). (Figure 1B). The adipose tissue gene expression of IRF5 correlated positively with clinical indicators of obesity including BMI (r = 0.37, P = 0.008) (Figure 1C) and body fat percentage (r = 0.51, P = 0.0004) (Figure 1D). HbA1c is a typical indicator of glycemic health in individuals with obesity and increasing HbA1c values indicate a rising risk for T2D, despite an overall reduction in blood pressure and cholesterol. Thus, progressive worsening of glycemic status is an independent predictor of disease progression. It is, therefore, important to assess the relationship between HbA1c and other putative marker(s) of metabolic inflammation and/or insulin resistance. In our obese individuals, IRF5 adipose expression was also found to be associated with HbA1c levels (r = 0.41, P = 0.02) (Figure 1E), whereas no such association was found in case of overweight population (r = 0.40, P = 0.09) (Appendix A). Overall, an agreement was found between gene and protein expression of IRF5 in the adipose tissue (r = 0.65, P = 0.001) (Figure 1F). Regarding other clinical markers, adipose IRF5 gene expression did not correlate with levels of fasting blood glucose (r = 0.03, P = 0.83), cholesterol (r = 0.40, P = 0.35), high-density lipoprotein (r = 0.10, P = 0.56), low-density lipoprotein (r = 0.18, P = 0.20), and triglycerides (r = 0.06, P = 0.70).

In addition to IHC, adipose IRF5 expression was also confirmed by confocal microscopy (CM) in lean, overweight, and obese individuals, five each. The representative IHC and CM images of IRF5 protein expression in the adipose tissue samples from lean, overweight, and obese are compared (Figure 2).

### 3.2. IRF5 Gene Expression Correlates with that of TNF-α but not IL-6 in Adipose Tissue 

Next, we asked if IRF5 expression changes in the adipose tissue were associated with adipose expression of TNF-α which is a signature proinflammatory cytokine. To this effect, our data show the increased TNF-α gene (P = 0.02, Figure 3A) and protein (P < 0.0001, Figure 3B) expression in obese subjects compared with lean. The representative IHC images from three independent determinations with similar results show TNF-α protein expression (arrows) in the adipose tissue samples from lean, overweight, and obese individuals (Figure 3C). A direct association was found between gene and protein expression of TNF-α in the adipose tissue (r = 0.51, P = 0.04) (Appendix A). Importantly, a strong positive correlation was found between adipose gene expression of IRF5 and TNF-α (r = 0.50, P = 0.0004) (Figure 3D).

We asked if IL-6 expression correlated with IRF5 expression in the adipose tissue. The data show that IL-6 gene (P = 0.03, Figure 4A) and protein (P < 0.0001, Figure 4B) expression was increased in the adipose tissue samples from obese individuals compared with lean. The representative IHC images from three independent determinations with similar results show IL-6 protein expression (arrows) in the adipose tissue samples from lean, overweight, and obese individuals (Figure 4C). A consensus was found between IL-6 gene and protein expression in the adipose tissue (r = 0.64, P = 0.004, Appendix A). However, the association between adipose gene expression of IRF5 and IL-6 could not attain statistical significance (r = 0.26, P = 0.07) (Figure 4D).

### 3.3. Adipose CXCL8 Expression is Enhanced in Obese Individuals and Associates with IRF5 Expression

We asked if the adipose CXCL8 and IRF5 expression correlated with each other. To this end, our data reveal that CXCL8 expression was elevated in obese compared with lean individuals, at both mRNA (P = 0.05, Figure 5A) and protein levels (P = 0.0001, Figure 5B). The representative IHC images from three independent determinations with similar results show CXCL8 protein expression (arrows) in the adipose tissue samples from lean, overweight, and obese individuals (Figure 5C). A direct association was found between gene and protein expression of CXCL8 in the adipose tissue (r = 0.50, P = 0.04, Appendix A). A positive correlation was found between adipose gene expression of IRF5 and CXCL8 (r = 0.40, P = 0.0004) (Figure 5D).

### 3.4. IRF5 Gene Expression Correlates with that of CCL5 but not CCL2 in Adipose Tissue

CCL5, also known as regulated on activation, normal T-cell expressed and secreted (RANTES), controls the chemotaxis of leukocytes including T lymphocytes, basophils, and eosinophils. We wanted to know if adipose CCL5 expression was elevated in obesity and whether these changes correlated with those of IRF5 mRNA expression in the adipose tissue. The data show that adipose CCL5 mRNA expression was relatively higher in obese individuals compared with lean (P = 0.11, Figure 6A), while CCL5 protein expression was significantly elevated in obese compared with lean individuals (P < 0.0001, Figure 6B). The representative IHC images from three independent determinations with similar results show the CCL5 protein expression (arrows) in the adipose tissue samples from lean, overweight, and obese individuals (Figure 6C). Similar to CCL2, the adipose CCL5 gene and protein expression did not associate with each other (r = 0.13, P = 0.54, Appendix A). Notably, an association was found between adipose gene expression of IRF5 and CCL5 (r = 0.40, P = 0.02) (Figure 6D).

CCL2, also known as monocyte chemoattractant protein (MCP)-1, is involved in recruiting monocytes, dendritic cells, and memory T cells to the sites of inflammation. We asked if CCL2 expression was elevated and correlated with IRF5 expression in the adipose tissue. The data show that adipose CCL2 gene (P = 0.05, Figure 7A) and protein (P < 0.0001, Figure 7B) expression was enhanced in obese individuals compared with lean. The representative IHC images from three independent determinations with similar results show CCL2 protein expression (arrows) in the adipose tissue samples from lean, overweight, and obese individuals (Figure 7C). CCL2 gene and protein expression in the adipose tissue did not associate with each other (r = 0.10, P = 0.70, Appendix A). Moreover, no correlation was found between IRF5 and CCL2 gene expression in the adipose tissue (r = 0.12, P = 0.42) (Figure 7D).

### 3.5. Relationship of IRF5 Gene Expression with Signature Inflammatory Immune Markers in the Adipose Tissue

The associations between adipose gene expression of IRF5 and other critical immune inflammatory markers including cytokines, chemokines, and chemokine receptors are summarized below in Table 2.

We and others have shown that the pattern recognition receptors are involved in metabolic inflammation. Therefore, we asked whether the elevated IRF5 expression had an association with TLRs expression. The data show a direct correlation between IRF5 gene expression and that of TLR2, TLR7, TLR8, and TLR9. IRF5 gene expression was also found to be associated with TLR-related signaling molecules including MyD88, IRAK-1, and IRF (Table 3).

The shift in macrophage polarization toward inflammatory M1 phenotype plays a pivotal role in the adipose tissue inflammation. Therefore, we asked if the adipose IRF5 gene expression correlated with inflammatory macrophage markers. Our data show strong positive associations between the adipose IRF5 gene expression and macrophage markers including CD11c, CD68, CD86, and CD163 (Table 4).

### 3.6. Adipose IRF5 Gene Expression Correlates with Systemic Immuno-Metabolic Markers

CRP is a well-known immune marker for systemic inflammation. Adiponectin is a metabolic marker which is a protein hormone involved in glucose regulation and fatty acid oxidation. We wanted to know if there was an association between the adipose IRF5 gene expression and these systemic immuno-metabolic markers. The data show that adipose IRF5 expression associated positively with CRP levels (Figure 8A) but negatively with adiponectin levels (Figure 8B).

The thematic illustration supporting the afore-mentioned data is shown below (Figure 9).

## 4. Discussion

Increase in the visceral adipose tissue mass has been correlated with increased inflammatory responses and a higher risk for the development of metabolic disease. The changes in the expression of interferon regulatory factors may have immunobiological consequences. In the present study, we report for the first time to our knowledge, that the adipose tissue expression of the transcription regulator IRF5 mRNA and protein was significantly higher in overweight and obese individuals compared with lean. The elevated expression of IRF5 was found to be associated positively with the clinical indicators of obesity such as BMI and body fat percentage. These changes in the adipose expression of IRF5 at gene and protein levels were found to be congruent. The transcription factor IRF5 is emerging as a new coplayer in the occurrence of obesity complications via its role as an orchestral conductor of macrophage polarization toward an inflammatory phenotype in the obese adipose tissue. This links the IRF5 directly to the nature and immunobiology of adipose tissue in obese individuals and to the related metabolic consequences. In line with this argument, a recent study shows that IRF5 knockdown in diet-induced obese mice led to the preferential growth of subcutaneous white adipose tissue as opposed to visceral white adipose tissue compared with wild-type mice [18]. Our IHC and confocal microscopy data support that obesity could play as a positive modulator of IRF5 expression in the subcutaneous adipose tissue in humans as is evident by a higher IRF5 expression in overweight and obese compared with lean individuals.

Adipose tissue, in addition to serving as an energy storage depot, also acts as an important endocrine organ that secretes a large number of cytokines, chemokines, and adipokines whose levels are modulated by obesity. At the initial stages of obesity, expanding adipose tissue becomes infiltrated by activated monocytes which differentiate into the resident macrophages, also called adipose tissue macrophages (ATMs). The ATMs are a predominant source of proinflammatory cytokines/chemokines in the adipose tissue. Our data show the elevated adipose expression of inflammatory cytokines including TNF-α, IL-1β, IL-6, and IL-18. TNF-α is a signature proinflammatory cytokine and increased adipose tissue expression of TNF-α was observed in obese mice [20]. TNF-α is mainly secreted by the activated monocytes/macrophages as well as other cells including lymphocytes, granulocytes, mast cells, endothelial cells, fibroblasts, and adipocytes. The increased expression of TNF-α in the adipose tissue associated with adiposity and insulin resistance [21,22]. IL-6 is a pleiotropic cytokine secreted by macrophages, T lymphocytes, and adipocytes. It can act as both a proinflammatory and an anti-inflammatory cytokine, depending on the nature of the tissue and the milieu. We and other reported the increased expression of IL-6/IL-6R in the adipose tissue in obesity [7,23,24]. Notably, increased IL-6 levels in obesity have been linked with metabolic inflammation, insulin resistance, and elevated circulatory CRP levels [25,26].

Chemokines are low molecular weight chemotactic cytokines that are induced by primary proinflammatory mediators and orchestrate the recruitment of well-defined leukocyte subsets into the site(s) of inflammation. Chemokines are classified based on the presence of two non-adjacent (called CXC chemokines or α-chemokines) or adjacent (called CC chemokines or β-chemokines) cysteine residues near the amino terminus. CXC chemokines are generally chemotactic for neutrophils while CC chemokines are chemoattractant for monocytes and a lymphocyte subset. Interestingly, we found the increased adipose gene expression of both α-chemokines (CXCL8, CXCL9, and CXCL10) and β-chemokines (CCL2, CCL5, and CCL7) in obese as compared to lean subjects. We speculate that the increased expression of these chemokines may play a role in promoting chronic inflammation via the recruitment of activated monocytes/macrophages in the adipose tissue. In support of this argument, TNF-α, IL-6, IL-8, CCL2, CCL4, CCL5, and CCL19 were found to be implicated with metabolic inflammation or insulin resistance in various tissues and organs [21,26,27,28,29,30,31,32]. Our data showing upregulated adipose expression of these chemokines in obese individuals are supported, at least in part, by other studies [28,33,34,35]. Our data also show a positive association of the adipose IRF5 gene expression with that of chemokine receptors such as CCR1, CCR2, and CCR5. Consistent with these data, IRF5 expression was found to be associated with various inflammatory markers [36,37,38,39,40,41,42]. However, we also noted discordance between gene and protein expression of CCL2 and CCL5 in the adipose tissue which may be due to variability in the stabilities of their mRNAs and proteins in the adipose tissue in obesity.

TLRs are emerging as key players in the inflammatory response that drives the expression of proinflammatory cytokines. We sought to determine the relationship between IRF5 expression in the adipose tissue and that of TLRs and related signaling molecules. Our data show strong positive correlations between adipose IRF5 gene expression and that of TLR2, TLR7, TLR8, TLR9, MyD88, IRAK-1, and IRF3 in obesity. Adipose IRF3 expression was found to be elevated in obese mice and humans and IRF3 knockdown prevented insulin resistance [43]. Similarly, upregulated expression of TLRs 1–9 and TLRs 11–13 was shown in two murine models of obesity [44]. We and others have also reported the increased expression of TLR2, TLR4, TLR7, TLR8, and TLR10 in obese and/or T2D individuals [19,45,46,47]. Taken together, these data suggest that the obesity-associated changes in adipose expression of IRF5 and TLRs are mutually congruent.

Changes in macrophage polarization, function, and metabolic signature are found in both inflammatory and metabolic diseases. In obesity and/or T2D, a typical proinflammatory M1 polarization in the adipose tissue is observed which is associated with certain morbid factors such as increased levels of TNF-α, fatty acids, glucose, insulin, and obesity-induced hypoxia [48,49]. Our data show strong positive associations between adipose gene expression of IRF5 and proinflammatory macrophage markers including CD11c, CD68, CD86, and CD163 whereas no association was found with the anti-inflammatory M2 macrophage marker CD302.

Furthermore, increased IRF5 expression in the adipose tissue was found to correlate positively with CRP levels and inversely with adiponectin levels in the circulation. CRP is the best characterized biomarker of systemic inflammation and it is an acute phase reactant protein synthesized in the liver in response to factors such as IL-6 which is secreted by activated macrophages and adipocytes [50]. Increased levels of CRP were found to correlate with metabolic inflammation, insulin resistance, adiposity, atherothrombosis, and other key features of metabolic syndrome [51]. Adiponectin is a protein hormone which is secreted by the adipose tissue and is involved in critical metabolic processes such as glucose regulation, insulin sensitivity, and fatty acid oxidation [52]. Increasing genetic and physiological evidence implicates the reduced adiponectin levels in obesity with insulin resistance and T2D [53,54] while, high adiponectin levels were found to associate with a lower risk for T2D development [55]. High adiponectin to CRP ratio is regarded as an indicator of the favorable metabolic and anthropometric profiles in obesity/T2D [56]. Together, these data suggest that the adipose tissue IRF5 gene expression in obese individuals, compared with lean, associates with a wide range of signature inflammatory markers including TNF-α, IL-1β, IL-6, IL-18, CXCL8, CXCL9, CXCL10, CCL2, CCL5, CCL7, and CRP. These data are also supported, at least in part, by previous studies in the context of obesity, inflammation, and insulin resistance [43,57,58,59,60,61,62,63].

Nonetheless, caution will be required while interpreting results of this preliminary study due to certain caveats, such as lack of data from the visceral adipose tissue and/or from the purified human adipocytes/preadipocytes. In addition, IHC could not be performed for all proinflammatory cytokines/chemokines or other inflammatory markers herein presented including TLRs, related signaling molecules as well as inflammatory macrophage markers due basically to the limited availability of the human adipose tissue biopsies. The caution will also be warranted while interpreting results of this preliminary study, especially as the lean cohort is very small and comprises of only six individuals. However, these concerns may be addressed during our future investigations in furthering this initial work.

In conclusion, our data show that the adipose tissue IRF5 gene/protein expression was significantly elevated in obesity which was found to be concordant with local and systemic inflammatory signatures, implicating that the IRF5 upregulation could be considered as a novel adipose marker for metabolic inflammation.

## Figures and Tables

**Figure 1 cells-08-01418-f001:**
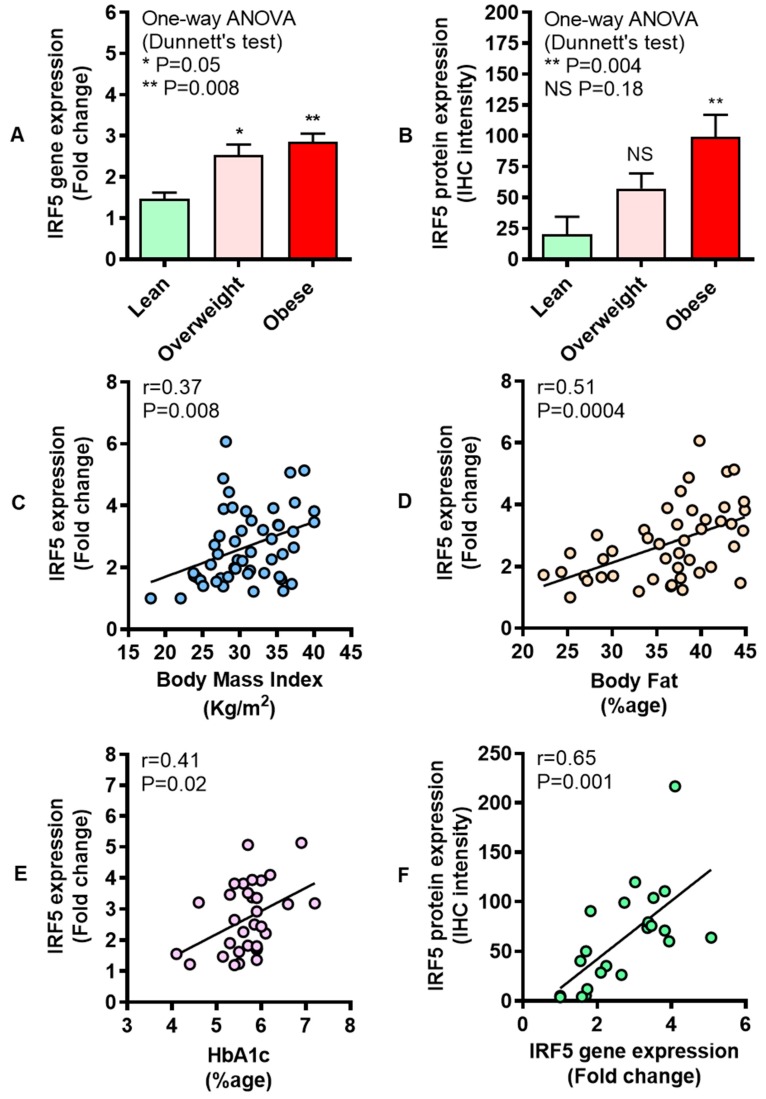
Enhanced adipose interferon regulatory factor-5 (IRF5) expression in obesity correlates with body mass index (BMI), body fat percentage, age, and levels of glycated hemoglobin (HbA1c). Adipose tissue IRF5 gene expression was determined in 53 non-diabetic individuals grouped as six lean, 18 overweight, and 29 obese using real-time RT-PCR. IRF5 protein expression was determined in six lean, eight overweight, and eight obese individuals using immunohistochemistry (IHC) as described in Materials and Methods. The data (mean ± SEM) show that (**A**) IRF5 adipose gene expression was elevated in overweight (P = 0.05) and obese (P = 0.008) compared with lean individuals. (**B**) As expected, IRF5 protein expression (arbitrary units) was also higher in obese (P = 0.004) compared with lean subjects; however, the difference between overweight and lean individuals could not attain statistical significance (P = 0.18). Furthermore, IRF5 adipose gene expression associated positively with clinical markers including (**C**) BMI (r = 0.37, P = 0.008), (**D**) body fat percentage, age (r = 0.51, P = 0.0004), and (**E**) HbA1c (r = 0.41, P = 0.02). (**F**) Adipose IRF gene and protein expression was found to be mutually concordant (r = 0.65, P = 0.001). The asterisks * and ** represent significance levels of P ≤ 0.05 and P < 0.01, respectively.

**Figure 2 cells-08-01418-f002:**
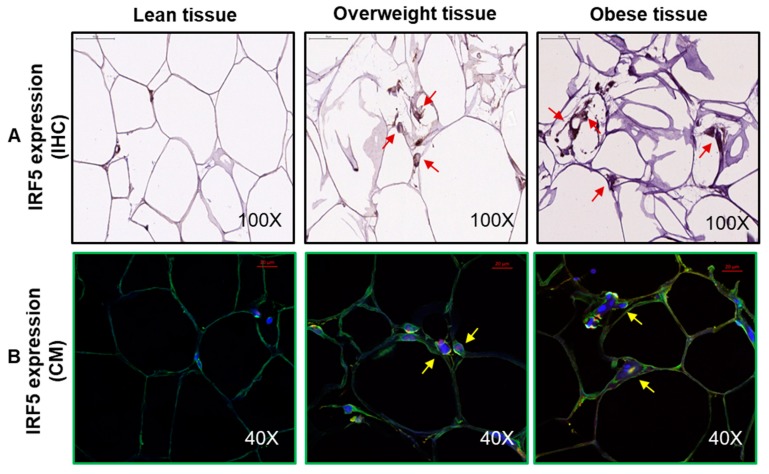
Increased interferon regulatory factor-5 (IRF5) protein expression in the adipose tissue. Adipose IRF5 protein expression was determined by immunohistochemistry (IHC) in six lean, eight overweight, and eight obese individuals and by confocal microscopy (CM), five individuals each, as described in Materials and Methods. The representative images obtained from five independent determinations with similar results show elevated adipose IRF5 protein expression in overweight and obese individuals compared with lean: (**A**) Red arrows in 100× magnification IHC images and (**B**) yellow arrows in 40× magnification CM images. Scale bar = 50 μM.

**Figure 3 cells-08-01418-f003:**
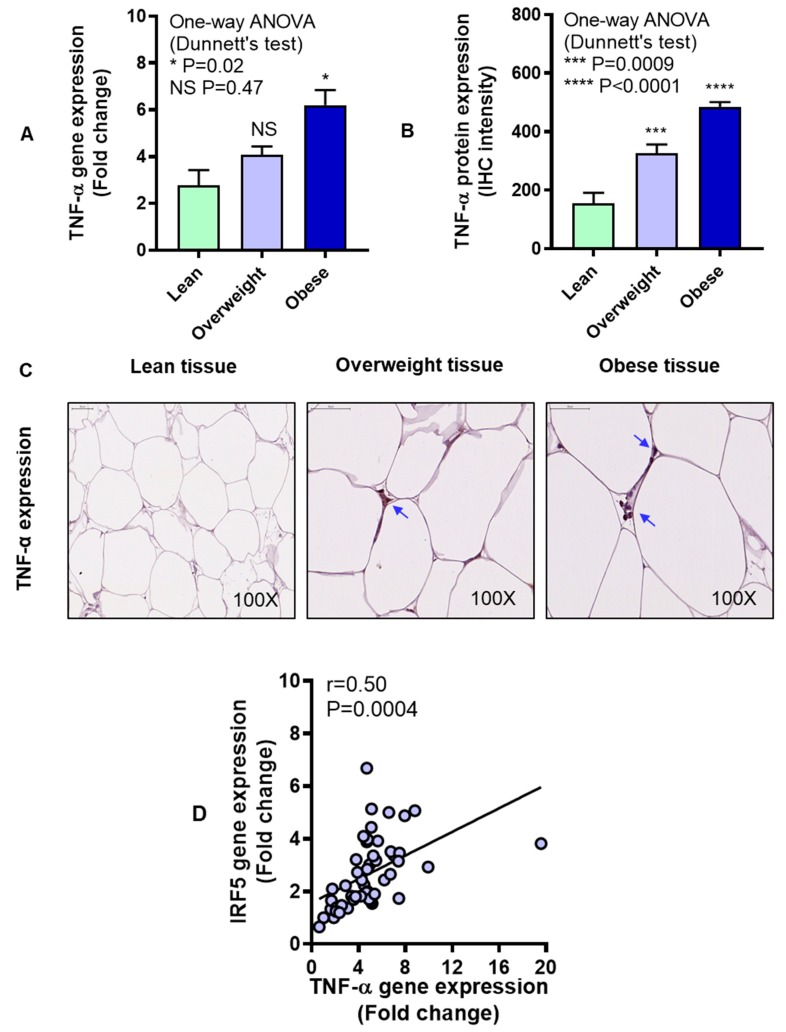
Increased adipose tumor necrosis factor-α (TNF-α) expression in obesity. Adipose TNF-α gene expression was determined in six lean, 18 overweight, and 27 obese non-diabetic individuals using real-time RT-PCR, while TNF-α protein expression was also determined in five lean, 13 overweight, and 10 obese non-diabetic individuals using immunohistochemistry (IHC) as described in Materials and Methods. The data (mean ± SEM) show that (**A**) TNF-α adipose gene expression was higher in obese (P = 0.02) compared with lean individuals. (**B**) TNF-α protein expression (expressed as IHC intensity in arbitrary units) was also elevated both in overweight (P = 0.0009) and obese (P < 0.0001) compared with lean individuals. (**C**) The representative IHC images obtained from three independent determinations with similar results show increased adipose tissue TNF-α expression (blue arrows, 100× magnification) in overweight and obese individuals compared with lean. Scale bar = 50 μM. (**D**) IRF5 and TNF-α gene expression correlated positively in the adipose tissue (r = 0.50, P = 0.0004). The asterisks *, ***, and **** represent significance levels of P ≤ 0.05, P < 0.001, and P < 0.0001, respectively.

**Figure 4 cells-08-01418-f004:**
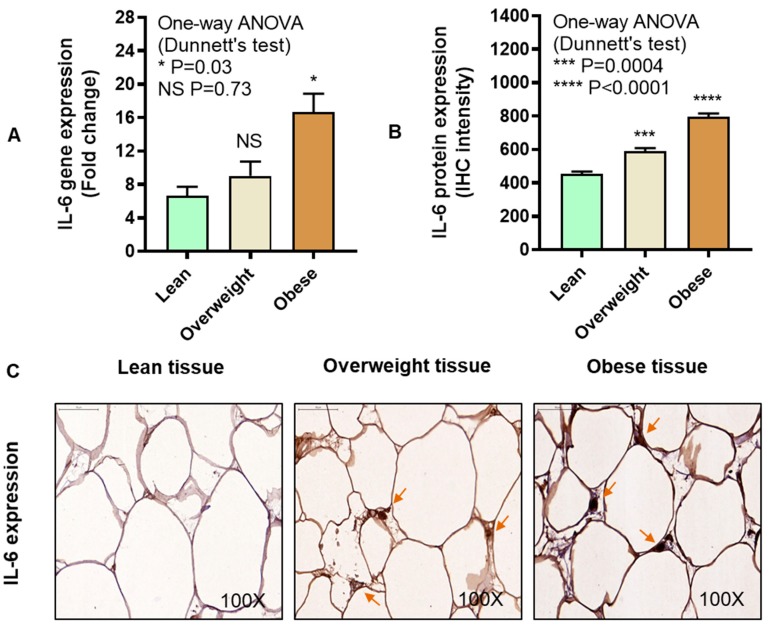
Elevated adipose interleukin-6 (IL-6) expression in obesity. Adipose tissue IL-6 gene expression was determined in five lean, six overweight, and 19 obese non-diabetic individuals using real-time RT-PCR, while IL-6 protein expression was also determined in six lean, 14 overweight, and 10 obese non-diabetic individuals using immunohistochemistry (IHC) as described in Materials and Methods. The data (mean ± SEM) show that (**A**) IL-6 adipose gene expression was elevated in obese (P = 0.03) compared with lean individuals. (**B**) IL-6 protein expression (IHC intensity expressed in arbitrary units) was also increased both in overweight (P = 0.0004) and obese (P < 0.0001) as compared with lean individuals. (**C**) The representative IHC images obtained from three independent determinations with similar results show elevated adipose tissue IL-6 expression (brown arrows, 100× magnification) in overweight and obese as compared with lean individuals. Scale bar = 50 μM. (**D**) IRF5 and IL-6 gene expression did not correlate in the adipose tissue (r = 0.26, P = 0.07). The asterisks *, ***, and **** represent significance levels of P ≤ 0.05, P < 0.001, and P < 0.0001, respectively.

**Figure 5 cells-08-01418-f005:**
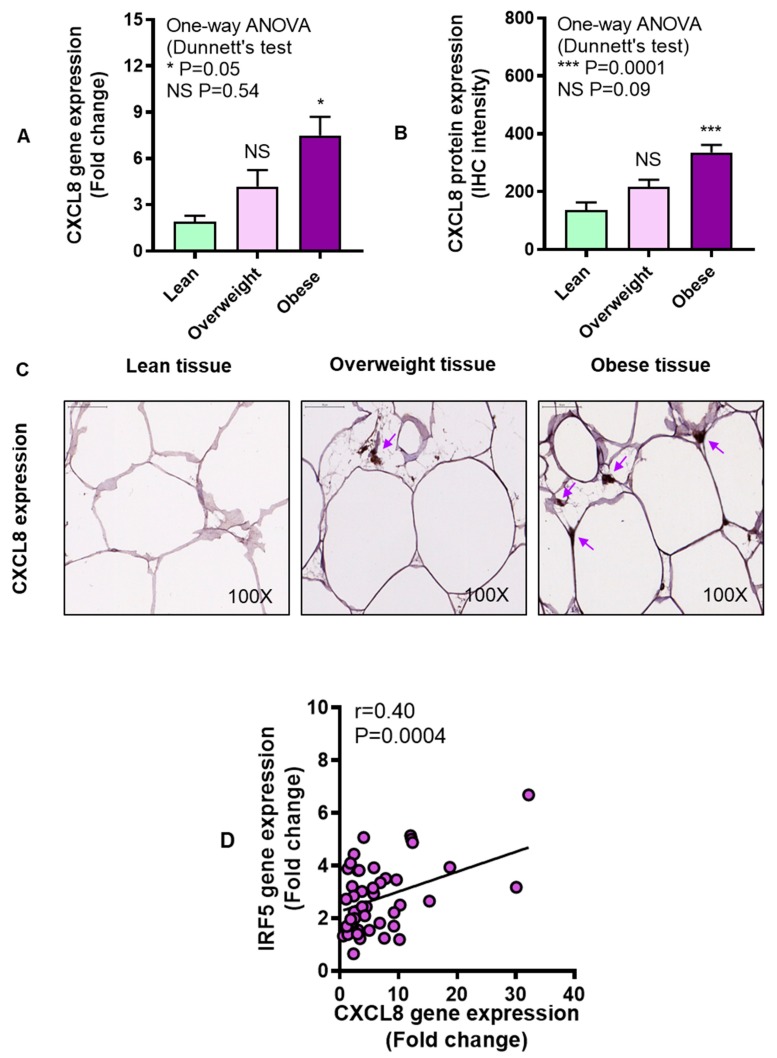
Upregulated adipose tissue chemokine CXC motif ligand-8 (CXCL8) expression in obesity. Adipose CXCL8 gene expression was determined in five lean, 18 overweight, and 27 obese non-diabetic individuals using real-time RT-PCR, while CXCL8 protein expression was determined in six lean, 14 overweight, and 10 obese non-diabetic individuals using immunohistochemistry (IHC) as described in Materials and Methods. The data (mean ± SEM) show that (**A**) adipose CXCL8 gene expression was upregulated in obese (P = 0.05) compared with lean individuals. (**B**) CXCL8 protein expression (IHC intensity in arbitrary units) was also higher in obese (P = 0.0001) compared with lean individuals. (**C**) The representative IHC images obtained from three independent determinations with similar results show upregulated adipose CXCL8 expression (purple arrows, 100× magnification) in overweight and obese as compared with lean individuals. Scale bar = 50 μM. (**D**) IRF5 and CXCL8 gene expression was associated with each other in the adipose tissue (r = 0.40, P = 0.0004). The asterisks * and *** represent significance levels of P ≤ 0.05 and P < 0.001, respectively.

**Figure 6 cells-08-01418-f006:**
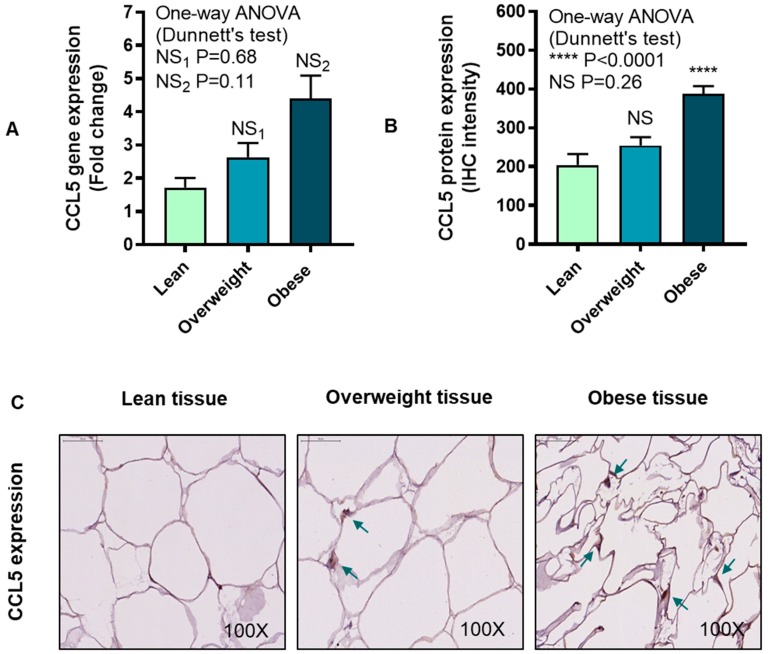
Adipose chemokine CC motif ligand-5 (CCL5) protein expression is elevated in obesity. CCL5 transcripts/messenger RNA (mRNA) expression was assessed in four lean, 18 overweight, and 23 obese non-diabetic subjects using real-time RT-PCR, while CCL5 protein expression was determined in six lean, 14 overweight, and 10 obese non-diabetic subjects using immunohistochemistry (IHC) as described in Materials and Methods. The data (mean ± SEM) show that (**A**) increase in CCL5 transcripts in overweight and obese compared with lean individuals was non-significant (P > 0.05). (**B**) CCL5 protein expression (IHC intensity shown in arbitrary units) was increased in obese (P < 0.0001) as compared with lean subjects. (**C**) The representative IHC images obtained from three independent determinations with similar results show increased CCL5 adipose expression (turquoise arrows, 100× magnification) in overweight and obese as compared with lean individuals. Scale bar = 50 μM. (**D**) IRF5 and CCL5 gene expression correlated in the adipose tissue (r = 0.40, P = 0.02). The **** asterisks represent significance level of P < 0.0001.

**Figure 7 cells-08-01418-f007:**
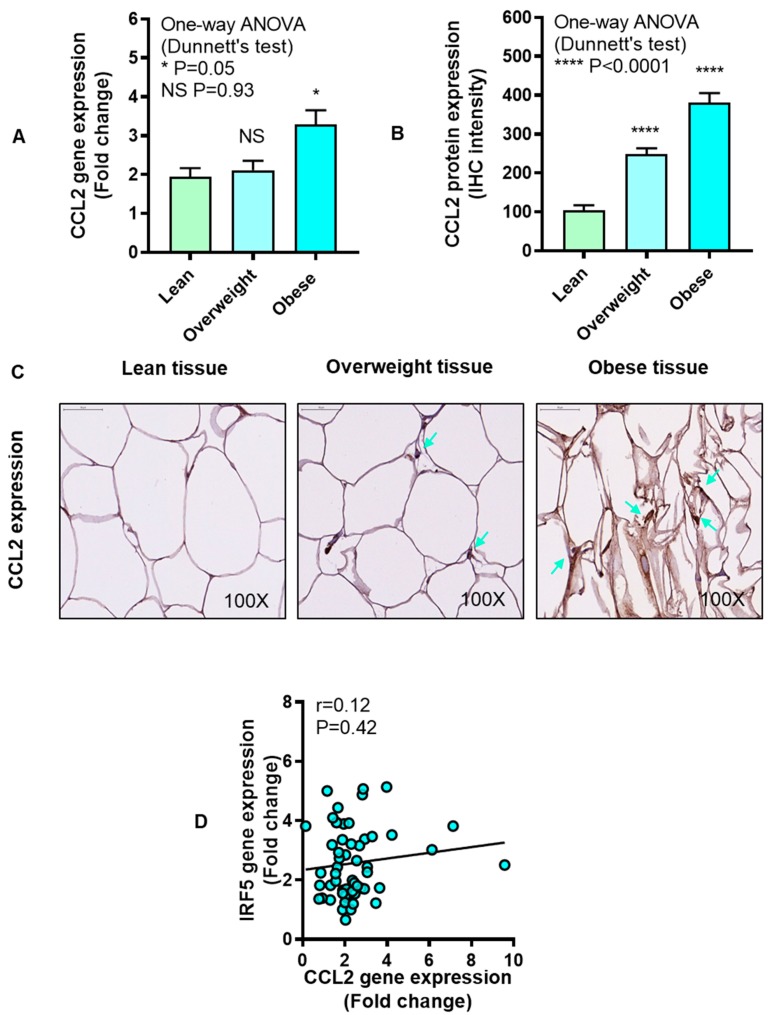
Increased adipose chemokine CC motif ligand-2 (CCL2) expression in obesity. CCL2 gene expression was determined in six lean, 18 overweight, and 23 obese non-diabetic individuals using real-time RT-PCR, while CCL2 protein expression was also assessed in six lean, 14 overweight, and 10 obese non-diabetic individuals using immunohistochemistry (IHC) as described in Materials and Methods. The data (mean ± SEM) show that (**A**) adipose CCL2 mRNA expression was upregulated in obese (P = 0.05) compared with lean subjects. (**B**) CCL2 protein expression (IHC intensity shown as arbitrary units) was also higher in overweight as well as obese individuals (P < 0.0001) as compared with lean. (**C**) The representative IHC images obtained from three independent determinations with similar results show elevated CCL2 adipose expression (cyan arrows, 100× magnification) in overweight and obese as compared with lean individuals. Scale bar = 50 μM. (**D**) There was no association found between IRF5 and CCL2 gene expression in the adipose tissue (r = 0.12, P = 0.42). The asterisks * and **** represent significance levels of P ≤ 0.05 and P < 0.0001, respectively.

**Figure 8 cells-08-01418-f008:**
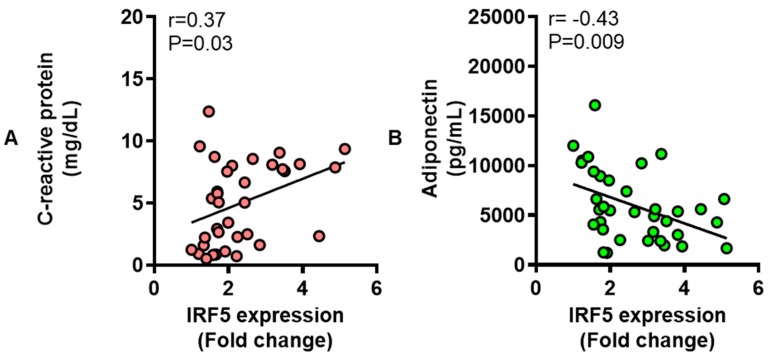
Adipose interferon regulatory factor-5 (IRF5) gene expression correlates with the circulatory C-reactive protein (CRP) and adiponectin levels. Plasma levels of CRP and adiponectin were measured in 35 individuals including lean, overweight, and obese by ELISA as described in Materials and Methods. The data (mean ± SEM) show that (**A**) adipose expression of IRF5 correlated positively with CRP levels (r = 0.37, P = 0.03) and (**B**) negatively with adiponectin levels (r = −0.43, P = 0.009).

**Figure 9 cells-08-01418-f009:**
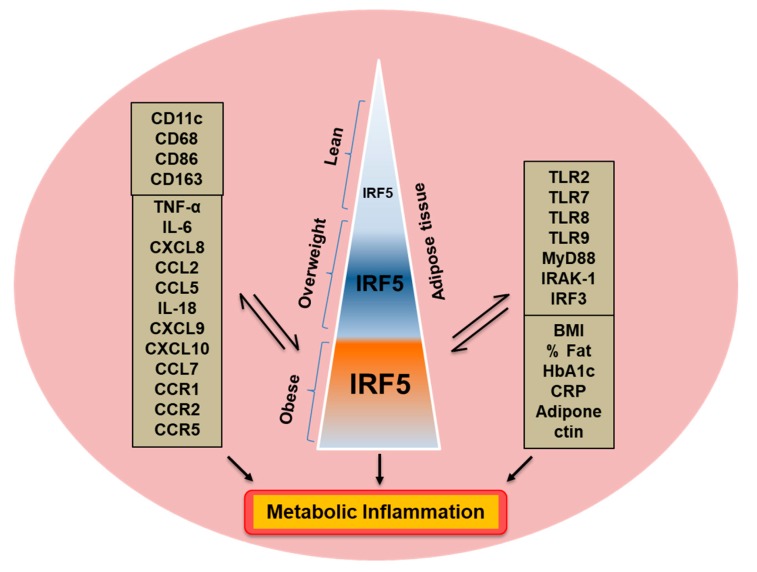
Increased adipose IRF5 expression concurs with metabolic inflammation. The illustration depicts a proposed model of metabolic inflammation, in support of the data presented, wherein the increase in obesity parallels with IRF5 adipose upregulation (positive associations with BMI, body fat percentage, age, and HbA1c). These changes in the IRF5 expression are in agreement with a wide range of markers of adipose inflammation including inflammatory cytokines/chemokines, TLRs and TLR-associated signaling molecules, inflammatory macrophage markers, as well as systemic immune-metabolic markers i.e., a positive association with circulatory CRP and a negative association with adiponectin levels. The two-way arrows (⇌) represent a mutual association. Taken together, these changes support the IRF5 upregulation as a novel marker for adipose tissue inflammation in obesity.

**Table 1 cells-08-01418-t001:** Patients’ clinical characteristics.

Total Number (N)	53 (24 Male and 29 Female)
Age (Yrs.)	43.04 ± 1.65 (Range: 24–71)
Subgroups (N)	6 Lean, 18 Overweight and 29 Obese
Body mass index (kg/m^2^)	31.32 ± 0.72
Body fat (%)	36.38 ± 0.89
Fasting glucose (mmol/L)	5.12 ± 0.07
HbA1c (%)	5.57 ± 0.06
Total cholesterol (mmol/L)	5.04 ± 0.14
HDL (mmol/L)	1.30 ± 0.05
LDL (mmol/L)	3.25 ± 0.13
Triglycerides (mmol/L)	1.12 ± 0.09
Hypertension (N)	4
Hyperlipidemia (N)	3
Therapy	Lipitor, Concor, Aldomet, Eltoxin

**Table 2 cells-08-01418-t002:** Correlation of adipose IRF5 gene expression with proinflammatory cytokines and chemokines/chemokine receptors.

Cytokines, Chemokines, and Chemokine Receptors	Correlation	Significance
IL-1β	r = 0.33	*P* = 0.04*
IL-18	r = 0.32	*P* = 0.04*
IL-23A	r = 0.16	*P* = 0.25
CXCL9	r = 0.28	*P* = 0.05*
CXCL10	r = 0.34	*P* = 0.02*
CCL7	r = 0.30	*P* = 0.04*
CCL11	r = 0.22	*P* = 0.14
CCL19	r = 0.20	*P* = 0.19
CCR1	r = 0.30	*P* = 0.05*
CCR2	r = 0.53	*P* = 0.0002***
CCR5	r = 0.44	*P* = 0.0006***

Note: Number of asterisks corresponds to level of statistical significance.

**Table 3 cells-08-01418-t003:** Correlation of adipose gene expression of interferon regulatory factor-5 (IRF5) with toll-like receptors (TLRs) and related transcription factors.

Receptor	Correlation	Significance
TLR2	r = 0.60	*P* < 0.0001****
TLR4	r = 0.001	*P* = 0.95
TLR7	r = 0.62	*P* < 0.0001****
TLR8	r = 0.55	*P* < 0.0001****
TLR9	r = 0.40	*P* = 0.005**
MyD88	r = 0.60	*P* < 0.0001****
IRAK-1	r = 0.49	*P* = 0.0004***
IRF3	r = 0.42	*P* = 0.005**

Note: Number of asterisks corresponds to level of statistical significance.

**Table 4 cells-08-01418-t004:** Correlation of adipose IRF5 gene expression with monocyte/macrophage markers.

Markers	Correlation	Significance
CD11c	r = 0.60	*P* < 0.0001****
CD68	r = 0.63	*P* < 0.0001****
CD86	r = 0.62	*P* < 0.0001****
CD163	r = 0.61	*P* < 0.0001****
CD302	r = 0.11	*P* = 0.43

Note: Number of asterisks corresponds to level of statistical significance.

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
