# Peer review of "Increased Adipose Tissue Expression of Interferon Regulatory Factor (IRF)-5 in Obesity: Association with Metabolic Inflammation"

_cells, 2019, doi:10.3390/cells8111418_

Round 1
Reviewer 1 Report
The paper is well planned; the resources used are very broad and provide a good overview about adipose tissue and inflammation.
Author Response
Response to Reviewer 1 Comments
Open Review
(x) I would not like to sign my review report
( ) I would like to sign my review report
English language and style
( ) Extensive editing of English language and style required
( ) Moderate English changes required
(x) English language and style are fine/minor spell check required
( ) I don't feel qualified to judge about the English language and style
|
Yes |
Can be improved |
Must be improved |
Not applicable |
|
|
Does the introduction provide sufficient background and include all relevant references? |
( ) |
(x) |
( ) |
( ) |
|
Is the research design appropriate? |
(x) |
( ) |
( ) |
( ) |
|
Are the methods adequately described? |
(x) |
( ) |
( ) |
( ) |
|
Are the results clearly presented? |
( ) |
(x) |
( ) |
( ) |
|
Are the conclusions supported by the results? |
(x) |
( ) |
( ) |
( ) |
Comments and Suggestions for Authors
The paper is well planned; the resources used are very broad and provide a good overview about adipose tissue and inflammation.
Submission Date
06 October 2019
Date of this review
15 Oct 2019 05:17:36
Author response to Reviewer’s general comments
English language style and minor spell check required: Done
Introduction, background and relevant references can be improved: Done (Please see page 2 of revised version)
Results section can be improved: Done (Please see pages 6-20 of revised version)
Reviewer 2 Report
Sindhu et al., investigated the correlation of IRF5 expression in fat tissue from lean, overweight and obese individuals with series of markers of metabolic inflammation. In particular, they checked if the level of IRF5 expression could be associated with (1) body weight, (2) inflammatory markers, (3) cytokines and (4) immune markers (macrophage M1).
They showed that IRF5 expression (both mRNA qPCR and protein IHC) in adipose tissue was significantly increased in obese individuals (not the overweighted through) and can be correlated with the level of TNFa as well as with the gene expression of CCL5 and various other inflammatory immune markers.
In conclusion, the authors identified IRF5 as a novel marker (both at the mRNA and protein level) of metabolic inflammation linked with obesity.
This is a descriptive work that is well assayed but the manuscript may benefit from some minor changes to avoid the catalogue-style of results.
The relevance to check the association with HbA1c should be briefly explained. Figure 3: Please show the correlation plot since this is the main novelty of these results (TNFa is a adipokine and its elevation in obese fat tissue is known). Same remark for all subsequent figures… I think it would be informative to see the correlative graphs since these are the main points that are discussed in this paper. Maybe merge paragraphs 3.2 and 3.3 in one result section? “IRF5 expression correlates with TNFa level but not IL6 in adipose tissue” Same remark for paragraphs 3.5 and 3.6 Figure 9: the figure suggests that IRF5 expression is upstream to many immune/inflammation markers. Since this work did not identify any causality relationships between IRF5 and these markers, I would suggest presenting these data in parallel instead of one in top of the other...(to avoid inferring causality).
Author Response
Response to Reviewer 2 Comments
Open Review
(x) I would not like to sign my review report
( ) I would like to sign my review report
English language and style
( ) Extensive editing of English language and style required
( ) Moderate English changes required
(x) English language and style are fine/minor spell check required
( ) I don't feel qualified to judge about the English language and style
|
Yes |
Can be improved |
Must be improved |
Not applicable |
|
|
Does the introduction provide sufficient background and include all relevant references? |
(x) |
( ) |
( ) |
( ) |
|
Is the research design appropriate? |
(x) |
( ) |
( ) |
( ) |
|
Are the methods adequately described? |
(x) |
( ) |
( ) |
( ) |
|
Are the results clearly presented? |
( ) |
(x) |
( ) |
( ) |
|
Are the conclusions supported by the results? |
(x) |
( ) |
( ) |
( ) |
Comments and Suggestions for Authors
The paper is well planned; the resources used are very broad and provide a good overview about adipose tissue and inflammation.
Submission Date
06 October 2019
Date of this review
15 Oct 2019 05:17:36
Author response to Reviewer’s general comments
English language style and minor spell check required: Done
Results section can be improved: Done (Please see pages 6-20 of revised version)
Author response to Reviewer’s specific comments
Sindhu et al., investigated the correlation of IRF5 expression in fat tissue from lean, overweight and obese individuals with series of markers of metabolic inflammation. In particular, they checked if the level of IRF5 expression could be associated with (1) body weight, (2) inflammatory markers, (3) cytokines and (4) immune markers (macrophage M1).
They showed that IRF5 expression (both mRNA qPCR and protein IHC) in adipose tissue was significantly increased in obese individuals (not the overweighted through) and can be correlated with the level of TNFa as well as with the gene expression of CCL5 and various other inflammatory immune markers.
In conclusion, the authors identified IRF5 as a novel marker (both at the mRNA and protein level) of metabolic inflammation linked with obesity.
This is a descriptive work that is well assayed, but the manuscript may benefit from some minor changes to avoid the catalogue-style of results.
Point 1: The relevance to check the association with HbA1c should be briefly explained.
Response 1: The relevance of HbA1c association is now explained as required. Please see the revised version of the MS, page 6, lines 177-181.
Point 2: Figure 3: Please show the correlation plot since this is the main novelty of these results (TNFa is a adipokine and its elevation in obese fat tissue is known). Same remark for all subsequent figures… I think it would be informative to see the correlative graphs since these are the main points that are discussed in this paper. Maybe merge paragraphs 3.2 and 3.3 in one result section? “IRF5 expression correlates with TNFa level but not IL6 in adipose tissue” Same remark for paragraphs 3.5 and 3.6.
Response 2: The reviewer’s comments are highly appreciated. In compliance of the same, correlation plots have now been added as Fig 3D, Fig 4D, Fig 5D, Fig 6D, and Fig 7D. Also, as suggested, sections 3.2 & 3.3 (pages 8-11) and 3.5 & 3.6 (pages 13-17) have been merged together.
Point 3: Figure 9: the figure suggests that IRF5 expression is upstream to many immune/inflammation markers. Since this work did not identify any causality relationships between IRF5 and these markers, I would suggest presenting these data in parallel instead of one in top of the other...(to avoid inferring causality).
Response 3: Fig 9 has been accordingly modified (Please see page 20 of the revised version).

Reviewer 3 Report
In this manuscript subcutaneous adipose tissue samples from 53 non-diabetic individuals (6 lean, 18 overweight and 29 obese) were investigated by RT-PCR to determine the gene expression of regulatory factor (IRF)-5 and a number of adipose inflammatory biomarkers. The IRF5 protein expression was also assessed using immunohistochemistry and/or confocal microscopy and systemic levels of C-reactive protein and adiponectin were measured by ELISA. The data of the present study show that IRF5 was significantly upregulated in obese compared with lean individuals and that IRF5 gene expression associated positively with adipose inflammatory biomarkers mentioned above as well as inflammatory macrophage markers. Thus, IRF5 upregulation may represent a novel adipose tissue marker for metabolic inflammation as elevated IRF5 expression in obese individuals concurs with the typical inflammatory signatures, locally and systemically.
The results of the study are interesting and the analysis of protein expression of adipose inflammatory markers are comprehensive and appears to be well-performed. However, one of my major concerns about this manuscript is the number of individuals included in the study. Although this is a preliminary study, the number of individuals are in my opinion too small to make any solid conclusions. In particular, the lean cohort is very small, as it constitute only of six individuals. The authors should therefore discuss the validity of the obtained data based on the small number of individuals from the different cohorts.
Furthermore, the introduction is very short and focus only on IRF5 but not the role of proinflammatory cytokines and chemokines in adipose inflammation. Consequently, the authors should elaborate more on the role of adipose inflammatory biomarkers in the introduction. Hence, the short background information on these biomarkers given in the Results section can then be deleted. References about the role of the individual inflammatory biomarkers in adipose inflammation is also lacking in the Results section and can then be included in the introduction.
Another concern is the ethical aspects of the study, which appears only to have been approved by the ethics committee of Dasman Diabetes Institute (DDI), Kuwait. The authors come from the same institute (DDI) and therefore this ethic committee cannot be considered as a completely independent ethical committee. In my country, it is a clear requirement that an independent ethic committee should approve clinical studies. Please explain.
In conclusion, an interesting study that needs some revision as described above before it can be accepted for publication in Cells.
Author Response
Response to Reviewer 3 Comments
Open Review
(x) I would not like to sign my review report
( ) I would like to sign my review report
English language and style
( ) Extensive editing of English language and style required
(x) Moderate English changes required
( ) English language and style are fine/minor spell check required
( ) I don't feel qualified to judge about the English language and style
|
Yes |
Can be improved |
Must be improved |
Not applicable |
|
|
Does the introduction provide sufficient background and include all relevant references? |
( ) |
( ) |
(x) |
( ) |
|
Is the research design appropriate? |
( ) |
(x) |
( ) |
( ) |
|
Are the methods adequately described? |
(x) |
( ) |
( ) |
( ) |
|
Are the results clearly presented? |
(x) |
( ) |
( ) |
( ) |
|
Are the conclusions supported by the results? |
(x) |
( ) |
( ) |
( ) |
Comments and Suggestions for Authors
The paper is well planned; the resources used are very broad and provide a good overview about adipose tissue and inflammation.
Submission Date
06 October 2019
Date of this review
15 Oct 2019 05:17:36
Author response to Reviewer’s specific comments
In this manuscript subcutaneous adipose tissue samples from 53 non-diabetic individuals (6 lean, 18 overweight and 29 obese) were investigated by RT-PCR to determine the gene expression of regulatory factor (IRF)-5 and a number of adipose inflammatory biomarkers. The IRF5 protein expression was also assessed using immunohistochemistry and/or confocal microscopy and systemic levels of C-reactive protein and adiponectin were measured by ELISA. The data of the present study show that IRF5 was significantly upregulated in obese compared with lean individuals and that IRF5 gene expression associated positively with adipose inflammatory biomarkers mentioned above as well as inflammatory macrophage markers. Thus, IRF5 upregulation may represent a novel adipose tissue marker for metabolic inflammation as elevated IRF5 expression in obese individuals concurs with the typical inflammatory signatures, locally and systemically.
Point 1: The results of the study are interesting, and the analysis of protein expression of adipose inflammatory markers are comprehensive and appears to be well-performed. However, one of my major concerns about this manuscript is the number of individuals included in the study. Although this is a preliminary study, the number of individuals are in my opinion too small to make any solid conclusions. In particular, the lean cohort is very small, as it constitutes only of six individuals. The authors should therefore discuss the validity of the obtained data based on the small number of individuals from the different cohorts.
Response 1: The reviewer’s comment is important. Accordingly, a cautionary note has been added at the end of the discussion section (Please see page 22, lines 483-485 of the revised version of the MS).
Point 2: Furthermore, the introduction is very short and focus only on IRF5 but not the role of proinflammatory cytokines and chemokines in adipose inflammation. Consequently, the authors should elaborate more on the role of adipose inflammatory biomarkers in the introduction. Hence, the short background information on these biomarkers given in the Results section can then be deleted. References about the role of the individual inflammatory biomarkers in adipose inflammation is also lacking in the Results section and can then be included in the introduction.
Response 2: The introduction has been now expanded by incorporating briefly about proinflammatory cytokines/chemokines that are critical as biomarkers of adipose inflammation. Please see page 2, lines 45-59 in the revised version of the MS. Accordingly, some relevant descriptions from the Results section have been removed/modified to avoid redundancy.
Point 3: Another concern is the ethical aspects of the study, which appears only to have been approved by the ethics committee of Dasman Diabetes Institute (DDI), Kuwait. The authors come from the same institute (DDI) and therefore this ethic committee cannot be considered as a completely independent ethical committee. In my country, it is a clear requirement that an independent ethic committee should approve clinical studies. Please explain.
In conclusion, an interesting study that needs some revision as described above before it can be accepted for publication in Cells.
Response 3: In addressing this comment, a few lines have been added to the relevant section as required. Please see page 2, lines 79-82.

Round 2
Reviewer 3 Report
The authors have carefully revised their manuscript according to my comments, except for the discussion on the validity of the obtained data based on the small number of individuals from the different cohorts. However, in the revised version of the manuscript, it is now mentioned that caution should be taken in the interpretation of the results of the study due to the very small number of individuals in the lean cohort. This satisfy to some extent my concerns about the interpretation of the results presented. Consequently, I find that the revised version of the manuscript is now acceptable for publication in Cells.